# *Valeriana rigida* Ruiz & Pav. Root Extract: A New Source of Caffeoylquinic Acids with Antioxidant and Aldose Reductase Inhibitory Activities

**DOI:** 10.3390/foods10051079

**Published:** 2021-05-13

**Authors:** Guanglei Zuo, Hyun-Yong Kim, Yanymee N. Guillen Quispe, Zhiqiang Wang, Kang-Hyuk Kim, Paul H. Gonzales Arce, Soon-Sung Lim

**Affiliations:** 1Department of Food Science and Nutrition, Hallym University, 1 Hallymdeahak-gil, Chuncheon 24252, Korea; B16504@hallym.ac.kr (G.Z.); khy9514@nate.com (H.-Y.K.); yany24@snu.ac.kr (Y.N.G.Q.); M20023@hallym.ac.kr (K.-H.K.); 2Department of Molecular Medicine and Biopharmaceutical Sciences, Graduate School of Convergence Science and Technology, Seoul National University, Seoul 151742, Korea; 3College of Public Health, Hebei University, Baoding 071002, China; wangzq2017@hbu.edu.cn; 4Laboratorio de Florística, Departamento de Dicotiledóneas, Museo de Historia Natural–Universidad Nacional Mayor de San Marcos, Avenida Arenales 1256, Lima 14-0434, Peru; pgonzalesarce@hotmail.com; 5Institute of Korean Nutrition, Hallym University, 1 Hallymdeahak-gil, Chuncheon 24252, Korea; 6Institute of Natural Medicine, Hallym University, 1 Hallymdeahak-gil, Chuncheon 24252, Korea

**Keywords:** *Valeriana rigida* Ruiz & Pav., antioxidant, aldose reductase, DPPH-HPLC, ultrafiltration-HPLC, HSCCC, pH-zone-refining CCC, caffeoylquinic acid, quantification

## Abstract

*Valeriana rigida* Ruiz & Pav. (*V. rigida*) has long been used as a herbal medicine in Peru; however, its phytochemicals and pharmacology need to be scientifically explored. In this study, we combined the offline 2,2-diphenyl-1-picryl-hydrazyl-hydrate (DPPH)-/ultrafiltration-high-performance liquid chromatography (HPLC) and high-speed counter-current chromatography (HSCCC)/pH-zone-refining counter-current chromatography (pH-zone-refining CCC) to screen and separate the antioxidants and aldose reductase (AR) inhibitors from the 70% MeOH extract of *V. rigida,* which exhibited remarkable antioxidant and AR inhibitory activities. Seven compounds were initially screened as target compounds exhibiting dual antioxidant and AR inhibitory activities using DPPH-/ultrafiltration-HPLC, which guided the subsequent pH-zone-refining CCC and HSCCC separations of these target compounds, namely 3-*O*-caffeoylquinic acid, 4-*O*-caffeoylquinic acid, 5-*O*-caffeoylquinic acid, 3,4-*O*-di-caffeoylquinic acid, 3,5-*O*-di-caffeoylquinic acid, 4,5-*O*-di-caffeoylquinic acid, and 3,4,5-*O*-tri-caffeoylquinic acid. These compounds are identified for the first time in *V. rigida* and exhibited remarkable antioxidant and AR inhibitory activities. The results demonstrate that the method established in this study can be used to efficiently screen and separate the antioxidants and AR inhibitors from natural products and, particularly, the root extract of *V. rigida* is a new source of caffeoylquinic acids with antioxidant and AR inhibitory activities, and it can be used as a potential functional food ingredient for diabetes.

## 1. Introduction

Natural products are important resources of antioxidants and aldose reductase (AR) inhibitors, which play an important role in preventing or treating diabetic complications [1,2,3,4]. As one of the most common chronic degenerative diseases, diabetes mellitus affected approximately 463 million people in 2019 [5], and a considerable number of diabetics might suffer from more than one diabetic complication [6,7]. Intensive glycemic control can delay the progression of diabetic complications, which may presumably occur due to the potential “metabolic memory” caused by early hyperglycemia and inadequate glycemic control [8]. Several pathways, such as polyol pathway, advanced glycation end products/receptors, hexosamine pathway, and others, can be activated and may reach the advanced stage in hyperglycemia condition; moreover, they may produce an excess of reactive oxygen species (ROS) and lead to cellular oxidative stress, which is considered an important cause and therapeutic target of diabetic complications [4,9,10,11]. Moreover, high blood glucose activates AR, the key enzyme in polyol pathway, and increases the polyol pathway flux in insulin-independent tissues, such as neural tissue, lens, retina, and kidney [3]. However, the increased polyol pathway flux may cause oxidative stress and sorbitol-induced osmotic stress in those tissues via decreasing the cytosolic NADPH/NADP^+^, increasing the cytosolic NADH/NAD^+^ ratios, and overproducing sorbitol, which subsequently contributes diabetic complications; accordingly, AR inhibitors hold the potential to ameliorate diabetic complications [3]. Notably, many natural components have revealed anti-diabetic potential via inhibiting ROS/oxidative stress and AR, such as curcuminoids, resveratrol [12], quercitrin [13], quercetin [14,15], and chlorogenic acid [16,17], which highlights the importance to screen and separate potent antioxidants and AR inhibitors from natural products.

*Valeriana rigida* Ruiz & Pav. (abbreviation in this study: *V. rigida*), or *Phyllactis rigida* (Ruiz & Pav.) Pers. (synonymous with *V. rigida*), is a perennial herb of the genus *Valeriana* L., which is native to Bolivia, Colombia, Ecuador, and Peru, and it has long been used as a traditional medicine in Peru [18,19,20,21]. Infusions of *V. rigida* or its mixtures with other herbs are used to treat headache, anxiety, insomnia, menopause, and nervous illness [19,21,22,23]. Nevertheless, little phytochemical information on *V. rigida* has been reported so far. Some species of *Valeriana* L., such as *V. officinalis* [24], *V. dioscoridis* [25], and *V. wallichii* [26] have previously shown anti-diabetic potential; however, to date no studies have linked *V. rigida* with anti-diabetic effects, in particular, antioxidants and AR inhibitory activities.

Targeting the bioactive compounds in the extract prior to separation and separating them in a target-guided manner can improve the separation efficiency and increase the “hit”. In particular, the antioxidants and enzyme inhibitors in extracts can be screened using 2,2-diphenyl-1-picrylhydrazyl (DPPH) radical reaction-based DPPH-high-performance liquid chromatography (HPLC) [27,28] and enzyme–ligand binding affinity-based ultrafiltration-HPLC [29,30,31], respectively. Moreover, as a versatile separation chromatography based on a continuous liquid–liquid partition, high-speed counter-current chromatography (HSCCC) with advantages of support matrix-free, no irreversible adsorption, low risk of sample denaturation, and high sample loading capacity has immense potential in separating natural products [32,33], and it is suitable to couple offline DPPH- and ultrafiltration-HPLC in order to achieve efficient screening and separation of bioactive compounds from natural product extracts [34,35,36]. Furthermore, pH-zone-refining counter-current chromatography (CCC), developed from conventional HSCCC, can be particularly used for the preparative separation of ionizable target compounds [32,37,38].

Therefore, the present study aimed to screen, separate, and identify the antioxidants and AR inhibitory components in the 70% MeOH root extract of *V. rigida* in a bioactivity-guided manner by coupling DPPH-HPLC, ultrafiltration-HPLC, pH-zone-refining CCC, and conventional HSCCC, followed by evaluation of antioxidant and AR inhibitory activities in vitro. Additionally, its main components were quantified using HPLC.

## 2. Materials and Methods

### 2.1. Reagents

The MeOH used for HPLC analysis was obtained from J. T. Baker (Avantor Performance Materials LLC, Center Valley, PA, USA), whereas the other organic solvents used for extraction and separation were obtained from Samchun Pure Chemical Co., LTD (Pyeongtaek, Korea). Dimethyl sulfoxide, sodium phosphate monobasic dihydrate, ammonium sulfate, sodium hydroxide, potassium phosphate monobasic, sodium phosphate dibasic dodecahydrate, *β*-nicotinamide adenine dinucleotide 2′-phosphate reduced tetrasodium salt hydrate (NADPH), ammonia solution (28–30%), formic acid (98%), glacial acetic acid, DL-glyceraldehyde (dimer), trifluoroacetic acid (TFA, 99%), Trolox, quercetin, 2,2′-azinobis (3-ethylbenzothiazoline-6-sulfonic acid) diammonium salt (ABTS), fluorescein sodium salt, 2,2′-azobis (2-methylpropionamidine) dihydrochloride (AAPH), 2,2-diphenyl-1-picrylhydrazyl (DPPH), sodium hypochlorite solution (10–15%), potassium persulfate, and 1,8-diaminonaphthalene (99%), were obtained from Sigma-Aldrich Chemical Co. (St. Louis, MO, USA). Human recombinant AR was purchased from Wako Pure Chemical Industries Ltd. (Osaka, Japan). Quercitrin was separated in our lab [39]. Epalrestat was purchased from CSNpharm (Arlington Heights, IL, USA). All the *n*-BuOH used for separations was saturated using water before use.

### 2.2. V. rigida Material and Preparation of Plant Extract

The dried root of *V**. rigida* was obtained from the department of La Libertad in Peru in a local market and preserved at the Center for Efficacy Assessment and Development of Functional Foods and Drugs, Hallym University. The specimen was authenticated by Paul H. Gonzales Arce from the Museo de Historia Natural, Universidad.

For preparation of the plant extract, the dried root of *V**. rigida* (70 g) was ground to powder and extracted using 2 L of 70% MeOH aqueous solution assisted by sonication for 4 h at room temperature (around 22 °C). The extraction was carried out twice, and the extraction solutions were combined, filtered (Advantec #2), and evaporated by rotary evaporation at 37 °C to gain about 16 g of extract powder.

### 2.3. HPLC Analysis

The chromatographic analyses were performed on a Dionex system (Sunnyvate, VA, USA) comprising a P850 pump, a STH585 column oven, ASI-100 auto-injector, and a UVD 170S detector. The separation was performed on a Capcell Park C18 UG120 column (5 μm, 4.6 mm id × 250 mm length; Shiseido Fine Chemicals, Tokyo, Japan) with the column temperature set up at 26 °C. Elution was carried out with a linear gradient of A (0.1% TFA) and B (MeOH) as follows: 0–5 min, 10–40% B; 5–14 min, 40–50% B; 14–18 min, 50–100% B; 18–24 min, 100% B; 24–26 min, 100–10% B; 26–30 min, 10% B. The samples were monitored at 254 nm and the injection volume used was 10–60 µL depending on the concentration of samples and experimental objects. 

### 2.4. Antioxidant Assay

#### 2.4.1. DPPH Radical Scavenging Assay

The experiment was carried out as previously described [39]. In brief, 180 µL of DPPH solution (in MeOH, 0.32 mM) and 20 µL of sample solution (in MeOH, extract 100–800 µg/mL, compounds 125–2000 µM) were mixed in a 96-well plate and allowed to react for 20 min at 25 °C in the darkness. Thereafter, the absorbance was measured using a microplate reader (EL800, Bio-Tek Instruments, Winooski, VT, USA) at 570 nm. Trolox, a reference antioxidant, was used to make a calibration curve derived from the DPPH radical scavenging activity (%) against the final concentrations of Trolox (6.25–100 µM), and the results (*n* ≥ 3) were presented as Trolox equivalent antioxidant capacity (TE, µM Trolox/µg extract or µM Trolox/µM compound). The inhibitory activity (%) of the samples against DPPH radical was calculated using Equation (1):(1)$$\%\;\mathrm{inhibition}=\left(1-\frac{A_{sample}-A_{blank1}}{A_{control}-A_{blank2}}\right)\times 100\%,$$where *A_control_* is the absorbance of DPPH solution free of samples, *A_sample_* is the absorbance of DPPH solution incubated with a sample, *A_blank_*_1_ is the absorbance of the test sample free of DPPH solution, and *A_blank_*_2_ is the absorbance of MeOH.

#### 2.4.2. ABTS Radical Scavenging Assay

The experiment was conducted as previously described [39]. In brief, 3.5 mM of potassium persulfate aqueous solution was used to prepare 0.2 mM of ABTS diammonium salt. The solution was diluted 10-fold using distilled water and allowed to produce ABTS radicals (ABTS^+^) by keeping the solution in the dark for 14 h at room temperature. Thereafter, 10 µL of sample (prepared in MeOH, extract 25–400 µg/mL, compounds 31.25–1000 µM) was mixed with 290 µL of ABTS^+^ solution in a 96-well plate and incubated in the dark for 10 min at 25 °C, which was followed by measuring the absorbance at 750 nm using the same microplate reader. Trolox, a reference antioxidant, was used to make a calibration curve derived from the ABTS^+^ scavenging activity (%) against the final concentrations of Trolox (1.04–16.67 µM), and the results (*n* ≥ 3) were presented as Trolox equivalent antioxidant capacity (TE, µM Trolox/µg extract or µM Trolox/µM compound). Equation (1) was used to calculate the ABTS^+^ scavenging activity (%), where *A_control_* is the absorbance of ABTS^+^ solution free of samples, *A_sample_* is the absorbance of ABTS^+^ solution incubated with a sample, *A_blank_*_1_ is the absorbance of the test sample free of ABTS^+^ solution, and *A_blank_*_2_ is the absorbance of the diluted potassium persulfate solution.

#### 2.4.3. Oxygen Radical Absorbance Capacity Assay

The oxygen radical absorbance capacity (ORAC) assay was performed as previously described [39]. Briefly, 0. 1 M PBS of pH 7.4 was used to prepare AAPH (40 mM) and fluorescein sodium salt (117 nm) shortly before use. Thereafter, 20 µL of sample (prepared in MeOH, extract 6.25–25 µg/mL, compounds 12.5–50) and 120 µL of fluorescein sodium salt (117 nm) were mixed and incubated for 15 min at 37 °C in a black 96-well plate. Next, 60 µL of AAPH (40 mM) was added to generate peroxyl radicals to initiate the reaction. The fluorescence intensity (*λ*_ex_ = 485 nm, *λ*_em_ = 538 nm) was monitored for 90 min (1 time per min) using a Fluoroskan Ascent FL microplate reader (Thermo, Waltham, MA, USA) maintained at 37 °C. Trolox was used to make a calibration curve derived from the net AUC (the area under the curve) values against the final concentrations of Trolox (1–10 µM), and the results (*n* ≥ 3) were expressed as Trolox equivalent antioxidant capacity (TE, µM Trolox/µg extract or µM Trolox/µM compound). The net AUC value was calculated by subtracting the AUC of the AAPH group (free of samples) from that of a sample group (with AAPH and a sample) using Equation (2):(2)$$\mathrm{AUC}=1+\sum_{n=0}^{n=90}f_{n}/f_{0},$$where *f_n_* is the fluorescence intensity measured at *n* min and *f*_0_ is the fluorescence intensity measured at 0 min.

#### 2.4.4. Hypochlorous Acid Scavenging Assay

The HOCl scavenging assay was adapted from a previous study using 1,8-diaminonaphthalene as a fluorescence probe for hypochlorite [40]. Briefly, 90 µL of 0.1 M PBS of pH 7.4, 20 µL of sample (prepared in water or MeOH aqueous solution, extract 62.5–250 µg/mL, compounds 62.5–1000 µM), and 40 µL of HOCl (10 µM in 0.1 M PBS of pH 7.4) were mixed and allowed to react for 5 min at room temperature in a black 96-well plate. Thereafter, 50 µL of fluorescence probe (1,8-diaminonaphthalene, 240 µM in distilled water) was added and allowed to react for 2 min. Next, the fluorescence intensity (*λ*_ex_ = 360 ± 20 nm, *λ*_em_ = 460 ± 20 nm) was immediately measured using a microplate fluorescence reader (FLx800; Bio-Tek, Winooski, VT, USA). Trolox (final concentration 25–100 µg/mL) was used as a reference antioxidant. Notably, DMSO remarkably interferes with the result, and therefore, should not be used in HOCl assay. The inhibitory activity (%) of the sample against HOCl was calculated using Equation (3) and presented as mean ± standard deviation (*n* = 3) and IC_50_ value (half-maximal inhibitory concentration), which was calculated using linear regression analysis: (3)$$\%\;\mathrm{inhibition}=\frac{f_{s}-f_{hp}}{f_{p}-f_{hp}}\times 100\%,$$
where *f_p_*, *f_hp_*, and *f_s_* are the fluorescence intensities of the probe alone (*f_p_*), the mixture of HOCl and probe (*f_hp_*), and the mixture of sample, HOCl, and probe (*f_s_*), respectively.

### 2.5. AR Inhibition Assay

The experiment was approved by the Institutional Animal Care and Use Committees (IACUC) of Hallym University (Hallym-2016-95). The eyes of 10-week Sprague–Dawley rats (250–280 g) were removed and frozen at −70 °C before use. Thereafter, the lenses were removed from the eyes, ground in a mortar (precooled at −70 °C), and extracted using 0.1 M PBS of pH 6.2 (around 0.5 mL of buffer per two frozen rat lenses). Then, the extraction solution was centrifuged using a 5417R centrifuge (Eppendorf, Germany) for 30 min at 10,000× *g* and 4 °C to get the rat lens AR homogenate (in the supernatant).

The AR inhibitory activities of the extract and the separated compounds were determined as described previously [39]. Briefly, 100 µL of 0.1 M PBS (pH 7.0), 20 µL of rat lens AR homogenate, 20 µL of NADPH (cofactor, 2.4 mM in 0.1 M PBS of pH 8.0), 20 µL of sample (extract 1.56–6.25 µg/mL, compounds 0.98–1000 µM in a mixture of water and DMSO), and 20 µL of ammonium sulfate solution (4 M in 0.1 M PBS of pH 7.0) were pipetted into a 96-well plate. Next, the reaction was initiated by adding 20 µL of the substrate (DL-glyceraldehyde dimmer, 25 mM in 0.1 M PBS of pH 7.0), and the values were further measured for 6 min at 340 nm (OD_340_) using an Epoch microplate spectrophotometer (BioTek Instruments, Winooski, VT, USA). Three compounds (final concentrations 0.31–250 µM) were used as positive controls, including two strong natural AR inhibitors, quercetin, and quercitrin [39], and one proved AR inhibitor drug, epalrestat [41]. The DMSO used for sample preparation was less than 0.4% (*v*/*v*) in the reaction system. The AR inhibitory activity (%) of the sample was calculated using Equation (4), and the results were presented as mean ± standard deviation (*n* = 3) and IC_50_ value, which was calculated using linear or logarithmic regression analysis depending on which one offered a better regression coefficient (*r*^2^):(4)$$\%\;\mathrm{inhibition}=\left(1-\frac{\left|Slope_{s}\right|-\left|Slope_{b}\right|}{\left|Slope_{c}\right|-\left|Slope_{b}\right|}\right)\times 100\%,$$where *Slope_b_, Slope_c_*, and *Slope_s_* are the slopes derived from the OD_340_ nm against the reaction time (min)—dotted lines of blank group (without enzyme or sample), the control group (without sample), and the sample group (with enzyme and sample), respectively. |*Slope*| is the absolute value of slope.

### 2.6. Screening of Antioxidants from the Extract Using Offline DPPH-HPLC

The offline DPPH-HPLC strategy was used to screen the potential antioxidants in the extract before separation. Briefly, 150 µL of DPPH solution (2.5 mg/mL in MeOH) and 50 µL of the extract solution (10 mg/mL in MeOH) were mixed and incubated for 30 min at 37 °C. Thereafter, the reaction solution (injection volume 10 µL) was subjected to HPLC assay. MeOH was used to replace the DPPH solution for incubation with the sample to be used as a DPPH-free control group. The compounds with reduced HPLC peak areas from DPPH group compared with those from the DPPH-free group were assigned as potential antioxidants.

### 2.7. Screening of AR Inhibitors from the Extract Using Ultrafiltration-HPLC

The enzyme–ligand binding affinity-based ultrafiltration was used to screen the potential AR inhibitors from the extract prior to separation. Briefly, 250 µL of 0.1 M phosphate-buffered saline (PBS, pH 7.0), 60 µL of human recombinant AR (0.05 units/mL in 0.1 M PBS of pH 6.2), and 20 µL of quercitrin (used as an enzyme blocker, 0.5 mg/mL) were mixed in a 1.5 mL tube and preincubated for 10 min at 37 °C. Thereafter, 40 µL of the extract (1 mg/mL in water) was added into the reaction mixture, and it was further incubated for 20 min at 37 °C. Next, the reaction mixture was ultrafiltrated through a Amicon^®®^ Ultra 10 kDa membrane (Merck Millipore Ltd., Tullagreen, Ireland) for 20 min at 13,000 rpm (10,770× *g*; Micro-12, Hanil Science Industrial Co., Incheon, South Korea) at 20 °C. Moreover, the filtrate was individually collected, and the centrifugal membrane was washed by adding 200 µL of 0.1 M PBS (pH 7.0) and further centrifuged for 20 min at 13,000 rpm. The two-time filtrates were combined and evaporated using nitrogen gas, which was then re-dissolved using 200 µL of 50% MeOH aqueous solution and subjected to HPLC assay with an injection volume of 60 µL. The AR and AR-free experiments were carried out simultaneously with and without AR, respectively, in the absence of quercitrin. Furthermore, the compounds with reduced HPLC peak areas in the AR group compared with those from the AR-free group and quercitrin-blocked AR group were assigned as AR inhibitors.

### 2.8. Separation of Target Compounds by pH-Zone-Refining CCC

The target compounds screened via ultrafiltration-HPLC were sensitive to the acid and base added into the CCC solvent systems, thereby indicating that these compounds are ionizable compounds and, therefore, they are suitable for separation via pH-zone-refining CCC [32].

#### 2.8.1. Screening of pH-Zone-Refining CCC Solvent System

The solvent system was screened via HPLC and evaluated according to the partition coefficient (*K* value) of the target compounds on the principle introduced by Ito [32]. Briefly, four solvent systems comprising EtOAc, *n*-BuOH, and H_2_O were first acidified by using formic acid to 208 mM or basified by using ammonia solution to 29 mM, and then the corresponding *K* values of the target compounds under acidic (*K*_acid_) or basic (*K*_base_) conditions were determined, as previously described [36]: each acidified or basified solvent system was partitioned to upper layer and lower layer. Thereafter, the extract (about 1–2 mg) was prepared in a 1.5 mL tube and dissolved by adding equal volumes (each 500 µL) of the upper layer and lower layers, which were mixed by a vortex mixer and centrifuged for about 20 s using a C1301 Mini Centrifuge (Labnet International, South Korea). Next, the upper and lower phase sample solutions were respectively pipetted (each 200 µL) into a new 1.5 mL-tube and evaporated by nitrogen gas. Each sample residue was redissolved using 200 µL of MeOH and subjected to HPLC analysis with an injection volume of 20 µL. The *K* value was calculated as *K* = *A_upper_*/*A_lower_*, where *A_upper_* and *A_lower_* are the HPLC peak areas of the target component in the upper and lower phases, respectively.

#### 2.8.2. Preparation of pH-Zone-Refining CCC Solvent System and Sample Solution

The solvent system EtOAc/*n*-BuOH/H_2_O (2:3:5, *v*/*v*) was selected as the solvent system for pH-zone refining CCC separation, which was prepared in a separating funnel and partitioned to upper and lower layers after equilibration. The upper layer was acidified by using formic acid (208 mM) as the stationary phase and the lower layer was basified by using ammonia solution (29 mM) as the mobile phase. Both the mobile and stationary phases were degassed by sonication before use. The sample solution was prepared by dissolving the extract (about 3.54 g) in the biphasic solvents containing 16 mL of the formic acid-acidified stationary phase and 3 mL of the ammonia-free mobile phase. Thereafter, an extra 208 mM of formic acid was added to the sample solution to further improve the solubility of the target compounds in the organic phase via the protonation effect. Next, the sample solution was centrifuged (UNION 32R PLUS; Hanil Science Industrial Co., Kimpo, Korea) for 10 min at 3720× *g* (4000 rpm) and only the supernatant was used for sample loading. 

#### 2.8.3. pH-Zone-Refining CCC Separation

The separation was performed on TBE 300C HSCCC equipment (Tauto Biotech. Co., Ltd., Shanghai, China) with three polytetrafluoroethylene multilayer coils (ID: 2.6 mm; total volume: 300 mL). A Biotage Isolera FLASH purification system (Uppsala, Sweden) was equipped with HSCCC equipment as a pump, a UV monitor, and an auto fraction collector. In brief, the stationary phase was first introduced to fill the HSCCC coil column at 50 mL/min, and then, the flow rate was set at 4 mL/min and the rotation speed of the coils was adjusted to 800 rpm. Thereafter, the sample solution was loaded, and the separation was initiated by introducing the mobile phase at 4 mL/min. The eluate was monitored and automatically collected by the Isolera FLASH purification system according to the changes of UV absorbance at 254 nm. Eventually, the stationary phase retention ratio was calculated as the volume of the stationary phase collected from the HSCCC coil column relative to the total volume of the HSCCC coil column after separation.

### 2.9. Separation of Target Compounds ***1*** and ***3*** by Conventional HSCCC

#### 2.9.1. Preparation of HSCCC Solvent System and Sample Solution

The solvent system *n*-BuOH/H_2_O (1:1, *v*/*v*) was prepared in a separating funnel and modified by adding acetic acid to 8.7 mM, followed by thorough mixing, and then partitioning to upper and lower phases after settling. The partitioned upper and lower phases were used as mobile and stationary phases, respectively, and were degassed by sonication before use. Approximately, 42 mg of the mixture of compounds **1** and **3** concentrated by pH-zone-refining CCC was dissolved in biphasic solvents comprising equal volumes (each 7 mL) of the mobile and stationary phases.

#### 2.9.2. HSCCC Separation

The stationary phase was first pumped to fill the HSCCC coil column; thereafter, the rotation rate of the HSCCC coil column was gradually regulated to 850 rpm. Next, the mobile phase was pumped in at 5 mL/min until a steady elution of the mobile phase from the column outlet line was observed. Then, the sample was loaded and eluted by the mobile phase at 5 mL/min and monitored at 254 nm. When the separation was complete, the stationary phase retention ratio was determined, as previously described.

### 2.10. Structure Identification

The structures of the compounds were identified by analyzing the data obtained from 400 MHz NMR (JNM-ECZ400S/L1; JEOL Ltd., Tokyo, Japan), 600 MHz NMR (Bruker Avance Neo 600 Ultra ShieldTM; Bruker Biospin, Germany), AB Sciex QTrap^®®^ 4500 LC/MS (Foster City, CA, USA), EI-MS (JEOL JMS-700; JEOL Ltd., Tokyo, Japan), and comparisons with references or standard compounds.

### 2.11. Quantification of the Major Compounds ***5***, ***6***, and ***8***

Although the three major compounds **5**, **6**, and **8** have different maximum UV absorbances, to simplify the quantification process, their contents in the extract were quantified using the same HPLC condition (injection volume 10 µL) described in Section 2.3. These three purified components (**5**, **6**, and **8**) were dissolved in MeOH at 1 mg/mL and diluted appropriately using MeOH to prepare standard solutions for making calibration curves and method validation. Calibration curves were obtained by plotting the HPLC peak areas (*y*) versus the corresponding concentrations (*x*, µg/mL) by triplicate injection of at least nine different concentrations of standard solutions 3,5-diCQA (**5**; 12.50–400.00 µg/mL), 4,5-diCQA (**6**; 12.50–400.00 µg/mL), and acacetin (**8**; 6.25–400.00 µg/mL). The limit of detection (LOD) and limit of quantification (LQD) were measured by signal-to-noise ratios of three (*S*/*N* = 3) and ten (*S*/*N* = 10), respectively. The repeatability and reproducibility of the quantification method were examined by measuring the relative standard deviation (RSD) values of the peak areas of each compound (50.00 and 200.00 µg/mL) determined by HPLC at intraday (*n* = 6) and interday (*n* = 3). The accuracy of the quantification method was examined by determining the spike recovery of each standard solution spiked in the extract solution. The standard solutions of 100.00 and 300.00 µg/mL and the extract of 1.00 mg/mL (in MeOH) were used for sample spiking by mixing 0.2 mL of the extract solution and 0.2 mL of each individual standard solution. The extract solution and the spiked solution were detected by HPLC in triplicate to calculate the spike recovery, as described previously [42], using Equation (5), and the content (µg/mg = mg/g) of each compound in the extract was calculated as the concentration (*C*_1_, µg/mL) from the corresponding calibration curve of each compound/extract concentration (1.00 mg/mL).
(5)$$\%\;\mathrm{Spike}\;\mathrm{recovery}=\frac{C_{2}\times V_{2}-C_{1}\times V_{1}}{C_{0}\times V_{0}}\times 100\%,$$
where *C*_0_, *C*_1_, and *C*_2_ are the concentrations of each compound tested in the standard solution, extract solution, and the spiked solution, respectively. *V*_0_, *V*_1_, and *V*_2_ are the volumes of the standard solution used for sample spiking, the extract solution used for sample spiking, and the spiked sample solution, respectively. In this study, *V*_0_ = *V*_1_ = 0.2 mL; *V*_2_ = *V*_0_ + *V*_1_ = 0.4 mL; *C*_1_ (µg/mL) and *C*_2_ (µg/mL) are calculated from the corresponding calibration curve of each compound, and *C*_0_ is the concentration of the standard solution (100.00 and 300.00 µg/mL) used for sample spiking. 

### 2.12. Statistical Analysis

The data obtained from DPPH and ABTS assays were analyzed using one-way analysis of variance (ANOVA) followed by a LSD’s multiple comparison Test (SPSS version 25; IBM, New York, NY, USA), whereas the data obtained from ORAC assay were analyzed using one-way ANOVA followed by a tamhane T2 Test (SPSS version 25), since the group variances are not equal (*F* = 0.03 < 0.05 by Levene’s test). A value of *p* < 0.05 was considered as statistically significant.

## 3. Results

### 3.1. Antioxidant and AR Inhibitory Activity of the 70% MeOH Root Extract of V. rigida

The 70% MeOH root extract of *V. rigida* exhibited comparable antioxidant activities toward Trolox against DPPH (0.75 µM Trolox equivalents/µg extract, TE), ABTS (0.82 TE), and peroxyl (2.60 TE) (ORAC assay) radicals (Appendix A). Moreover, the extract was able to scavenge HOCl radicals (IC_50_ 16.52 µg/mL), although it was weaker than Trolox (IC_50_ 8.52 µg/mL) (Table 1). Notably, the AR inhibitory activity of the extract (IC_50_ 0.478 µg/mL) was remarkably higher than quercetin (IC_50_ 4.536 µg/mL), which is a popular natural AR inhibitor with anti-diabetic potential [15]; however, the extract was less active than quercitrin (IC_50_ 0.046 µg/mL), which is one of the most active natural AR inhibitors [13], and epalrestat (IC_50_ 0.016 µg/mL), one proved AR inhibitor drug (Table 1). In general, the extract exhibited remarkable antioxidant and AR inhibitory activities, thereby suggesting that the components it contains may be used as new sources of antioxidants and AR inhibitors with health-promoting benefits including anti-diabetic properties.

### 3.2. Screening of Antioxidants and AR Inhibitors from the Extract Using Offline DPPH- and Ultrafiltration-HPLC

To screen the antioxidants and AR inhibitors from the extract prior to separation, offline DPPH-HPLC and ultrafiltration-HPLC were performed. As illustrated in Figure 1, after reaction with DPPH radicals, a significant reduction in the HPLC peak areas of compounds **1**–**7** was found in DPPH group compared with that of DPPH-free group; therefore, these seven compounds were screened as antioxidants in the 70% MeOH extract of *V. rigida* root via DPPH-HPLC method.

The AR inhibitors in the extract were screened using the ultrafiltration-HPLC based on enzyme-ligand binding affinity. Potential AR inhibitors would bind to the active site of AR, forming macromolecular enzyme–ligand complexes during incubation with AR, and thus, they cannot pass through the ultrafiltration membrane via centrifugation, thereby leading to decreased concentration of the AR inhibitors in the filtrate of the AR-containing group (AR group) compared with those from the AR-free group. Thereafter, the resulting concentration difference was determined by analyzing the centrifugal filtrates of AR and AR-free groups using HPLC as Section 2.7 described, by which compounds **1**–**7** were preliminarily screened as potential inhibitors since the HPLC peak areas of these components in the centrifugal filtrate of AR group reduced compared with those in the AR-free group (Figure 2B). Moreover, considering that false positives could arise from nonspecific binding of compounds to AR nonfunctional sites, AR was preincubated with quercitrin, a strong AR inhibitor, to block the active site of AR, thereby reducing the possibility of other AR inhibitors from binding to the AR active site. As illustrated in Figure 2C, blocking AR with quercitrin (quercitrin-blocked AR group) resulted in an increase in the HPLC peak areas of AR inhibitors in the centrifugal filtrate compared with those from the AR group (free of quercitrin), thus verifying compounds **1**–**7** as AR inhibitors. Consequently, by DPPH-HPLC and ultrafiltration-HPLC methods, compounds **1**–**7** were screened as target components with dual antioxidant and AR inhibitory activities, which could guide further HSCCC separation.

### 3.3. Selection of Solvent System and Separation of Target Compounds by pH-Zone-Refining CCC and Conventional HSCCC

A successful HSCCC separation mainly depends on the selection of a suitable solvent system, which is expected to satisfy the *K* values of the target compounds with 0.5–2.0, and the separation factor (*α*) of two objective compounds higher than 1.5 (*α* = *K*_a_*/K*_b_ ≥ 1.5, *K*_a_ ≥ *K*_b_) [43]. Several solvent systems consisting of *n*-hexane, EtOAc, MeOH, and H_2_O in different proportions were initially tested. As indicated in Appendix A, multiple runs of conventional HSCCC using several solvent systems would be required to separate all the target compounds, since their *K* values could not be covered by a single HSCCC solvent system (0.5 ≤ *K* ≤ 2.0); however, this would be time-consuming, require more solvent, and laborious. Thereafter, we tried to modify the solvent system *n*-hexane/EtOAc/MeOH/H_2_O (0.2:5:1.8:5, *v*/*v*) by adding a little formic acid. Surprisingly, the *K* values of target compounds markedly increased, indicating that these compounds are acid compounds since their solubility toward the organic phase significantly increased after protonation by formic acid (Appendix A). Accordingly, we decided to separate the target compounds using pH-zone-refining CCC, which is particularly suitable for separating ionizable analytes [43]. 

As for pH-zone-refining CCC, a good solvent system is expected to satisfy *K* ≫ 1 under acidic condition (*K*_acid_ ≫ 1) and *K* ≪ 1 under basic condition (*K*_base_ ≪ 1) for acid compounds [32]. We first modified the solvent systems EtOAc/H_2_O (1:1, *v*/*v*), EtOAc/*n*-BuOH/H_2_O (4:1:5, *v*/*v*), EtOAc/*n*-BuOH/H_2_O (3:2:5, *v*/*v*), and EtOAc/*n*-BuOH/H_2_O (2:3:5, *v*/*v*) with 208 mM formic acid and with 30 mM ammonia, respectively, and then, the *K*_acid_ and *K*_base_ values of the target compounds were determined using the modified solvent systems. As listed in Table 2, the *K*_acid_ values of the target compounds offered by the 208 mM formic acid-acidified EtOAc/*n*-BuOH/H_2_O (2:3:5, *v*/*v*) solvent system were greater than those obtained from other solvent systems, whereas the *K*_acid_ values of the target compounds offered by the 30 mM ammonia-basified EtOAc/*n*-BuOH/H_2_O (2:3:5, *v*/*v*) solvent system were significantly lower than 1, thereby indicating that EtOAc/*n*-BuOH/H_2_O (2:3:5, *v*/*v*) is a suitable pH-zone-refining CCC solvent system for separation of the target compounds. As described in Section 2.8.2, the solvent system EtOAc/*n*-BuOH/H_2_O (2:3:5, *v*/*v*) was prepared in a separating funnel and divided into upper and lower phases. The upper phase was acidified using formic acid at a final concentration of 208 mM to be used as the mobile phase, and the lower phase was basified using ammonia at a final concentration of 29 mM to be used as the stationary phase. Moreover, formic acid retains the target compounds (acid compounds) in the stationary phase (organic phase) via the protonation effect, whereas ammonia elutes the target compounds (acid compounds) in the mobile phase (aqueous phase) through the deprotonation effect.

Furthermore, the target compounds were separated as described in Section 2.8.3. As illustrated in Figure 3, by a single run of pH-zone-refining CCC, target compounds **1** (55.7 mg), **2** (57.8 mg), **4** (59.5 mg), **5** (146.5 mg), **6** (300.2 mg), and **7** (56.0 mg) were separated from the centrifugal supernatant of the sample solution (about 3.54 g of sample was used to prepare sample solution) with purities over than 93% by HPLC assay at 254 nm; however, target compound **3** was eluted as a mixture of **3** and **1** (42.0 mg). After separation, the stationary phase retained in the CCC coil column was collected to calculate the stationary phase retention ratio (about 30%). Notably, the major nontarget compound **8** was found to precipitate in the form of crystals in the collected stationary phase during storage in the hood. The remaining solvent system was carefully discarded, and then, the crystals were washed out using MeOH and evaporated to obtain high-purity compound **8** (25.7 mg, Figure 3B). 

The mixed compounds **1** and **3** were further separated using conventional HSCCC. As listed in Table 3, the 8.7 mM acetic acid-modified *n*-BuOH/H_2_O (1:1, *v*/*v*) provided satisfactory *K* values and *α* value for target compounds **1** and **3** (*K*_1_ = 0.73, *K*_3_ = 1.57, *α* = *K*_3_*/K*_1_ = 2.15 > 1.5) and was therefore used for HSCCC separation of compounds **1** and **3**. The separation was conducted as described in Section 2.9.2. The target compounds **1** (10.8 mg) and **3** (13.1 mg) were successfully separated from the mixture of compounds **1** and **3** (about 42 mg) (Figure 4). After the separation was complete, the final volume retention ratio of the stationary phase was determined to be 63%.

### 3.4. Identification of the Separated Compounds

The molecular weights (MW) of the compounds were determined by LC-ESI-MS (negative ion) (target compounds **1**–**7**) or EI-MS (compound **8**) as follows: compounds **1**–**3**, MW 254, LC-ESI-MS [M-H]^−^ 253; compounds **4**–**6**, MW 516, LC-ESI-MS [M-H]^−^ 515; compound **7**, MW 678, LC-ESI-MS [M-H]^−^ 677; compound **8**, MW 284, EI-MS parent ion [M]^+^ 284. By analyzing the ^1^H-NMR (Appendix A), LC-ESI-MS (negative ion), EI-MS data, and comparison of the data with published references, the compounds **1**–**8** were identified as 3-*O*-caffeoylquinic acid (**1**) [44], 5-*O*-caffeoylquinic acid (chlorogenic acid, **2**) [44], 4-*O*-caffeoylquinic acid (**3**) [44], 3,4-di-*O*-caffeoylquinic acid (**4**) [44], 3,5-di-*O*-caffeoylquinic acid (**5**) [44], 4,5-di-*O*-caffeoylquinic acid (**6**) [44], 3,4,5-tri-*O*-caffeoylquinic acid (**7**) [45], and acacetin (**8**) [46]. The structures are illustrated in Figure 5. Notably, all these components are reported for the first time in *V. rigida*. Moreover, according to the results obtained from offline DPPH-HPLC and ultrafiltration-HPLC analysis, the seven caffeoylquinic acids (target compounds **1–7**) are deduced to be the main antioxidants and AR inhibitors in the 70% MeOH root extract of *V. rigida.*

### 3.5. Antioxidant and AR Inhibitory Activity of the Target Compounds ***1***–***7***

The antioxidant and AR inhibitory activities of the target compounds **1**–**7** were further verified using DPPH, ABTS, ORAC, and HOCl radical scavenging assays (Table 4 and Table 5) and AR inhibition assay (Table 5).

Overall, the target compounds **1**–**7** exhibited comparable antioxidant activities to Trolox but varied in DPPH, ABTS, ORAC, and HOCl assays (Table 4 and Table 5). As summarized in Table 4, the DPPH radical scavenging activity of the compounds followed the order 3,4,5-*O*-tri-caffeoylquinic acid (3,4,5-triCQA), 4,5-diCQA > 3,4-diCQA > Trolox > 3,5-diCQA, 4-CQA > 5-CQA > 3-CQA, which was consistent with that obtained from ABTS radical scavenging assay. Nevertheless, this result differed from that of the ORAC (Table 4) and HOCl scavenging (Table 5) assays. In ORAC assay, all the target compounds exhibited higher antioxidant activity than Trolox as follows: 4,5-diCQA, 4-CQA, 3,4-diCQA > 5-CQA, 3,5-diCQA, 3,4,5-triCQA > 3-CQA > Trolox (Table 2). In HOCl assay (Table 5), based on the assessed IC_50_ values, the HOCl scavenging activities of the target compounds followed the descending order: 3,5-diCQA > 4,5-diCQA > 3,4-diCQA > 3,4,5-triCQA > Trolox > 4-CQA > 5-CQA > 3-CQA.

Moreover, all the target compounds (**1**–**7**) exhibited remarkable AR inhibitory activities with IC_50_ values ranging from 0.151 to 7.80 µM, which were more active than quercetin (IC_50_ 15.01 µM) but less active than quercitrin (IC_50_ 0.102 µM) and epalrestat (IC_50_ 0.049 µM) (Table 5). Among the caffeoylquinic acids tested, 3,5-diCQA was the most active AR inhibitor, followed by 3,4-diCQA, 4,5-diCQA, 3,4,5-triCQA, 5-CQA, 4-CQA, and 3-CQA; whereas the nontarget compound, acacetin (**8**), revealed weak AR inhibitory activity, even at 100 µM (inhibition ratio 36.3%). The results proved that the antioxidant and AR inhibitory activities of the 70% MeOH root extract of *V. rigida* are mainly contributed by the caffeoylquinic acids, and the efficacy of DPPH-HPLC and ultrafiltration-HPLC as antioxidants and AR inhibitors screening methods prior to separation were verified.

### 3.6. Quantification of the Major Compounds ***5***, ***6***, and ***8***

The three main compounds 3,5-diCQA (**5**), 4,5-diCQA (**6**), and acacetin (**8**) in the 70% MeOH root extract of *V. rigida* were quantified using HPLC as described in Section 2.11. The HPLC method was validated by checking the linearity, LOD, LOQ, repeatability, reproducibility, and accuracy (spike test). As summarized in Table 6, all these three compounds exhibited good linearity (r^2^ > 0.99) within the concentrations tested (12.5–400 µg/mL for **5** and **6**, 6.25–400 µg/mL for **8**). The detection sensitivity of **5** and **6** was higher than **8** as indicated by their LOD (0.05–0.06 µg/mL for **5** and **6**, 0.20 µg/mL for **8**) and LOQ (0.17–0.20 µg/mL for **5** and **6**, 0.65 µg/mL for **8**) (Table 6). The established method was proven to have reliable repeatability and reproducibility by intraday testing (RSD 5.80%–8.70% for **5**, **6**, and **8** at 50.00 µg/mL and 200.00 µg/mL) and interday testing (RSD 3.30%–7.40% for **5**, **6**, and **8** at 50.00 µg/mL and 200.00 µg/mL) (Table 6).

The spike recovery of these three compounds was within 96.00–101.21% (Table 7). These results suggested that the quantification method established was reliable and accurate for quantifying the content of compounds **5**, **6**, and **8** in the extract. As summarized in Table 7, the 70% MeOH root extract of *V. rigida* was proved to contain high contents of **5** (66.47 mg 3,5-diCQA/g extract), **6** (112.95 mg 4,5-diCQA/g extract), and **8** (23.26 mg acacetin/g extract).

## 4. Discussion

The present study is the first to demonstrate that the 70% MeOH root extract of *V. rigida* is a new source of caffeoylquinic acids with antioxidant and AR inhibitory activities. Several species of *valeriana* L. genus have long been used as traditional medicine including *V. rigida* [20,22]. Some of these have been extensively studied for their phytochemicals and pharmacological properties, such as *V. officinalis* L. [47], *V. jatamansi* Jones [48,49], *V. officinalis var. latifolia* [50], *V. wallichii* [51], *V. edulis* [52], *V. spp.* [53], *V. fauriei* [54], *V. dioscoridis* [25], etc.

Our results revealed that the 70% MeOH root extract of *V. rigida* possesses remarkable in vitro antioxidant and AR inhibitory activities (Appendix A and Table 1), which are mainly attributed to the presence of seven caffeoylquinic acids, 3-CQA, 4-CQA, 5-CQA, 3,4-diCQA, 3,5-diCQA, 4,5-diCQA, and 3,4,5-triCQA as demonstrated by offline DPPH-/ultrafiltration-HPLC (Figure 1 and Figure 2) and 96-well plate assays (Table 4 and Table 5). Moreover, the contents (mg compound/g extract) of the major components 3,5-diCQA (**5**), 4,5-di-CQA (**6**), and acacetin (**8**) were 66.47 mg/g, 112.95 mg/g, and 23.26 mg/g, respectively (Table 7). Among the compounds tested, 4-CQA was found to have higher activity than 5-CQA and 3-CQA against DPPH, ABTS, peroxyl (ORAC assay), and HOCl radicals (Table 4 and Table 5), thereby supporting the fact that the esterification of caffeoyl group at the C-4 position of quinic acid is more relevant for antiradical activity [55]; whereas 3,4-diCQA and 4,5-diCQA appeared to exert higher activity than 3,5-diCQA against DPPH, ABTS, and peroxyl (ORAC assay) radicals (Table 4). This is in accordance with the previous study reporting that adjacent 4,5-diCQA and 3,4-diCQA exhibit higher activity than the nonadjacent diCQAS against DPPH, ABTS, and PTIO radicals [56]. Furthermore, 3,4,5-triCQA exhibited higher activity than diCQAs (4,5-diCQA, 3,4-diCQA, and 3,5-diCQA) (Table 4), which was followed by mono CQAs (4-CQA, 5CQA, and 3-CQA) against DPPH and ABTS radicals, whereas 3,4,5-triCQA exhibited lower peroxyl and HOCl radical scavenging activity than the diCQAs tested (Table 4 and Table 5), which may be attributed to the potential steric hindrance effect caused by an increasing number of caffeoyl groups. In addition to the radicals used in this study, caffeoylquinic acids also inhibit lipid peroxidation [57], reactive oxygen, and nitrogen species [58]; however, for the first time, we compared the scavenging activity of these caffeoylquinic acids toward HOCl radical (Table 5), which is a strong ROS that can be generated by activated phagocytes and may be implicated in the pathogenesis of various diseases, including cardiovascular disease [59], Alzheimer disease [60], and diabetes [61]. Moreover, these caffeoylquinic acids possess remarkable AR inhibitory activity (Table 5), which is much higher than that of quercetin, a popular natural AR inhibitor with anti-diabetic activity [15]. Among the caffeoylquinic acids tested in our study, diCQAs (3,5-diCQA, 3,4-diCQA, and 4,5-diCQA) exhibited higher AR inhibitory activity than 3,4,5-triCQA, followed by mono CQAs (5-CQA, 4-CQA, and 3-CQA); however, these results are inconsistent with a previous study reporting that 3,4,5-triCQA is more active than diCQAs [62]. This may be caused by the difference in the experimental conditions and may require further study. Notably, antioxidants and AR inhibitors hold therapeutic potential in diabetic complications [3,4,9], and numerous natural compounds have the potential to ameliorate diabetic complications by inhibiting oxidative stress and AR [12,14,63,64,65]. Particularly, 5-CQA (chlorogenic acid) is able to prevent diabetic nephropathy by inhibiting oxidative stress and inflammation in diabetic rats [17] and prevent cataractogenesis by inhibiting AR in galactose-fed rats [16]. Moreover, the consumption of caffeoylquinic acids-rich coffee is associated with reduced type 2 diabetes risk in clinical trials [66]. Collectively, from the data obtained in this study and other references, the caffeoylquinic acids-rich root extract of *V. rigida* has the potential to be used as an anti-diabetic ingredient in functional foods or medicines; therefore, further toxicity and animal studies are warranted for verifying its safety and anti-diabetic effects.

In addition, the present study suggests that the combination of offline DPPH-, ultrafiltration-HPLC, and HSCCC/pH-zone-refining CCC can be used for efficient screening and separation of antioxidants and AR inhibitors from natural products. Compounds **1**–**7** were screened as target compounds with dual antioxidant and AR inhibitory activities using DPPH-HPLC and ultrafiltration-HPLC (Figure 1 and Figure 2), and their activities were further confirmed by antioxidant and AR inhibition assays (Table 4 and Table 5). In contrast, the nontarget compound **8** exhibited very weak antioxidant [67] and AR inhibitory activities in this study (Table 5), thus proving that as antioxidants and AR inhibitor screening tools, offline DPPH-HPLC and ultrafiltration-HPLC can improve the hit. Chemical reaction-based DPPH offline has been widely used to couple with HPLC [27,28] or even GC [68] for screening antioxidants, which also functioned well in this study, whereas AR-ligand binding affinity based ultrafiltration-HPLC has been less used [69,70,71]; however, no competitive experiments were carried out for those ultrafiltration-HPLC studies, which may lead to nonspecific binding of compounds with enzymes or centrifugal membrane, causing false positives [30,72]. The inactivation of AR by heating results in loss of its binding ability to AR inhibitors, which has been recently used to verify the screening result by comparing it with that obtained from the active AR group [73]. Nevertheless, the enzyme precipitates in the solution after heat deactivation in our test, thereby affecting the screening result by blocking the centrifugal membrane, as described previously [72]. Our result proves that the preincubation of AR with quercitrin, one of the most active natural AR inhibitors, can block the active site of AR, reducing the possibility of other AR inhibitors from binding to AR active site (Figure 2). This suggests that introducing a quercitrin-blocked AR in the control group for verification can be used to reduce the false positives caused by nonspecific binding in AR ultrafiltration-HPLC assay, as those have been carried out in other enzymes [30,72].

As for the separation of caffeoylquinic acids, Zhang et al. previously separated 3,4,5-triCQA (12.7 mg), 1,3,5-triCQA (15.2 mg), and 3-CQA (42.5 mg) from the raw material of *Hypericum perforatum* (10.02 g) by HSCCC using solvent EtOAc/MeOH/H_2_O (5:2:5, *v*/*v*) [74]. Tong et al. succeeded in the preparative separation of 3,5-diCQA (0.289 g), 3,4-diCQA (0.106 g), and 3-CQA (0.090 g) from the sample of *Lonicerae* flos (2.136 g) by pH-zone-refining CCC using MtBE/CAN/H_2_O (2:2:3, *v*/*v*) as the basic solvent system (retainer 10 mM TFA; eluter 8 mM NH_4_OH) [75]. More recently, Liu et al. separated 1,3-diCQA, 1-CQA, and 3-CQA by two-step HSCCC using *n*-Hexane/EtOAc/MeOH/H_2_O (1:5:2.5:5 + 0.05% formic acid, *v*/*v*) and *n*-BuOH/EtOH/saturated (NH_4_)₂SO₄/H_2_O (9:1:6:4, *v*/*v*) as solvent systems. However, using the separation method established in this study, in a single run of pH-zone-refining CCC, six caffeoylquinic acids including 3-CQA (55.7 mg), 5-CQA (57.8 mg), 3,4-diCQA (59.5 mg), 3,5-diCQA (146.5 mg), 4,5-diCQA (300.2 mg), and 3,4,5-triCQA (56.0 mg) were separated from the 70% MeOH root extract of *V. rigida* (≈3.54 g) (Figure 3). Therefore, the pH-zone-refining CCC separation method established in this study is presumed to be an important reference for the preparative separation of caffeoylquinic acids.

## 5. Conclusions

In conclusion, the present study is the first to report that the 70% MeOH root extract of *V. rigida* possesses considerable in vitro antioxidant and AR inhibitory activities, and moreover, it demonstrated that the antioxidant and AR inhibitory activities of the extract were mainly attributed to the presence of seven caffeoylquinic acids 3-CQA (**1**), 4-CQA (**3**), 5-CQA (**2**), 3,4-di-CQA (**4**), 3,5-diCQA (**5**), 4,5-diCQA (**6**), and 3,4,5-triCQA (**7**) using offline DPPH-/ultrafiltration-HPLC analysis and DPPH, ABTS, ORAC, HOCl, and AR inhibition assays. The results indicated that the caffeoylquinic acids-rich root extract of *V. rigida* can act as a potential functional food or medicine ingredient for diabetes; therefore, further animal studies may be required to further assess its activity. Moreover, the results also suggested that the offline DPPH-/ultrafiltration-HPLC and HSCCC/pH-zone-refining CCC can be combined to efficiently screen and separate antioxidants and AR inhibitors from natural products.

## Figures and Tables

**Figure 1 foods-10-01079-f001:**
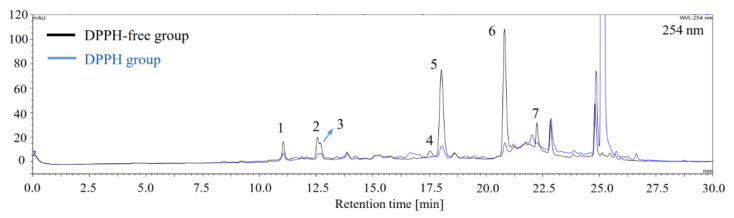
Chromatogram of DPPH-HPLC for screening antioxidants from the 70% MeOH extract of *V. rigida* root. Compounds **1**–**7** were identified as antioxidants via DPPH-HPLC, since the HPLC peak areas (254 nm) of these seven components reduced after reaction with DPPH radicals (DPPH group) compared with those from the DPPH-free group.

**Figure 2 foods-10-01079-f002:**
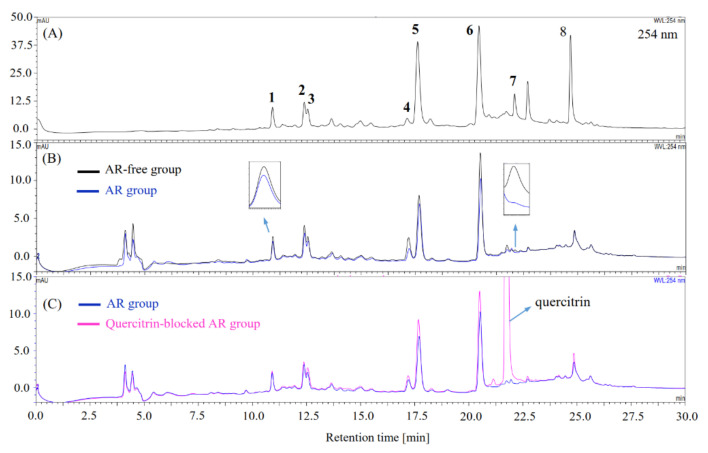
Chromatogram of ultrafiltration-HPLC for screening aldose reductase (AR) inhibitors from the 70% MeOH extract of *V. rigida* root. (**A**) HPLC chromatography (254 nm) of the 70% MeOH extract of *V. rigida* root; (**B**) chromatograms of the centrifugal filtrates of AR-free group (black line) and AR group (blue line); (**C**) chromatograms of the centrifugal filtrates of AR group (blue line) and quercitrin-blocked AR group (red line). Compounds **1**–**7** were screened as AR inhibitors.

**Figure 3 foods-10-01079-f003:**
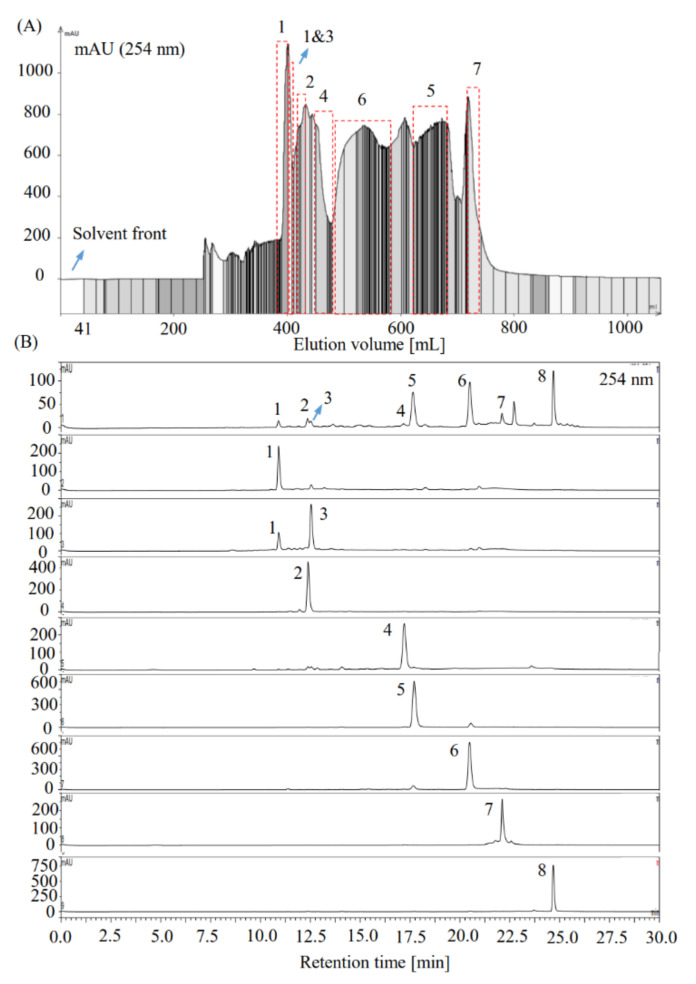
Chromatograms of pH-zone-refining CCC separation and HPLC detection of the target compounds from the 70% MeOH extract of *V. rigida* root. (**A**) pH-zone-refining CCC separation of the target compounds **1–7** from the extract using solvent system EtOAc/*n*-BuOH/H_2_O (2:3:5, *v*/*v*). The upper phase of EtOAc/*n*-BuOH/H_2_O (2:3:5, *v*/*v*) was acidified using formic acid (208 mM) as the stationary phase, whereas the lower phase of EtOAc/*n*-BuOH/H_2_O (2:3:5, *v*/*v*) was basified using ammonia (30 mM) as the mobile phase. Revolution speed: 800 rpm; mobile phase flow rate: 4 mL/min; UV detection wavelength 254 nm. (**B**) HPLC chromatograms (254 nm) of the 70% MeOH extract of *V. rigida* root and the compounds separated from it by pH-zone-refining CCC. Notably, compound **8** was obtained by its natural crystallization in the remaining stationary phase collected after separation.

**Figure 4 foods-10-01079-f004:**
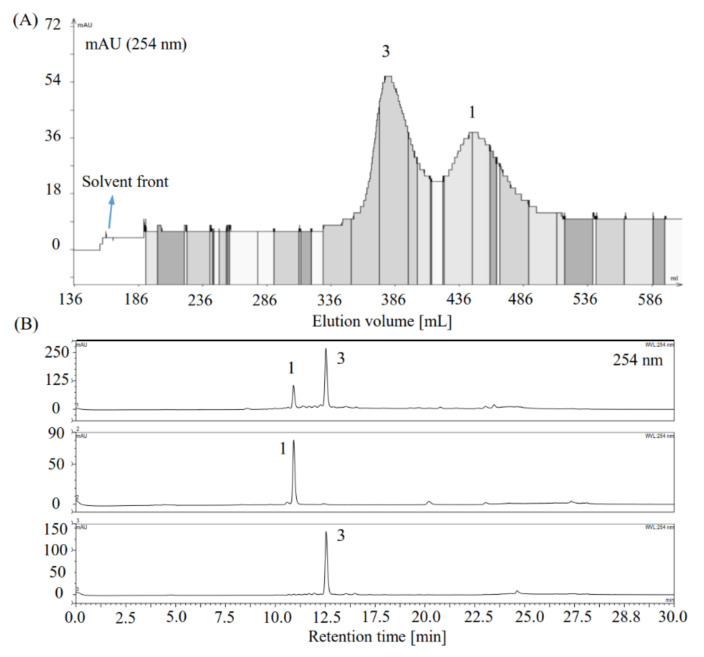
Chromatograms of conventional HCCC separation and HPLC detection of the target compounds **1** and **3**. (**A**) HSCCC separation of the target compounds **1** and **3** from their mixture using the 8.7 mM acetic acid-modified *n*-BuOH/H_2_O (1:1, *v*/*v*) solvent system. The partitioned upper and lower phases of the 8.7 mM acetic acid-modified *n*-BuOH/H_2_O (1:1, *v*/*v*) solvent system were used as the mobile and stationary phases, respectively. Revolution speed 850 rpm; mobile phase flow rate: 5 mL/min; UV detection wavelength 254 nm. (**B**) HPLC chromatograms (254 nm) of the mixture of compounds **1** and **3** and the separated compounds by HSCCC.

**Figure 5 foods-10-01079-f005:**
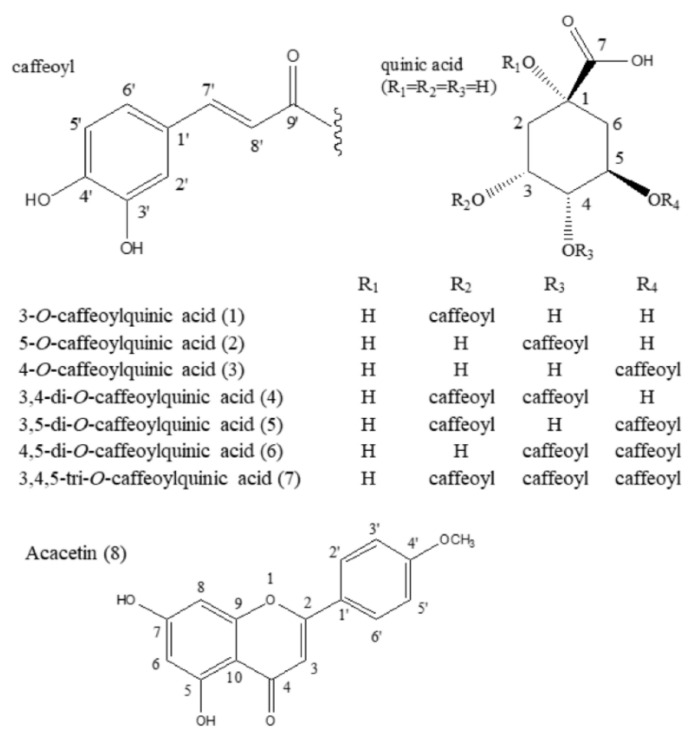
The structures of the components separated from the 70% MeOH root extract of *V. rigida*.

**Table 1 foods-10-01079-t001:** Hypochlorous acid scavenging and AR inhibitory activity of the 70% MeOH root extract of *V. rigida.*

Sample *^a^*	HOCl Scavenging Activity			AR Inhibitory Activity		
	Concentration (µg/mL)	Inhibition (%) *^b^*	IC_50_ *^c^* (µg/mL)	Concentration (µg/mL)	Inhibition (%)	IC_50_ (µg/mL)
Extract	25	67.00 ± 2.65	16.52	0.625	61.45 ± 2.34 *^b^*	0.478
	12.5	42.74 ± 1.58		0.313	36.25 ± 0.63	
	6.25	28.38 ± 2.25		0.156	26.47 ± 3.02	
Trolox	25	97.07 ± 5.45	8.52	-	-	-
	12.5	66.53 ± 2.54		-	-	
	6.25	40.76 ± 0.39		-	-	
Quercetin	- *^d^*	-	-	7.556	67.05 ± 3.22	4.536
	-	-		3.778	45.98 ± 4.49	
	-	-		1.889	34.78 ± 0.51	
Quercitrin	-	-	-	0.112	74.57 ± 1.11	0.046
	-	-		0.056	62.33 ± 0.44	
	-	-		0.028	32.57 ± 2.70	
Epalrestat	-	-	-	0.040	91.81 ± 1.84	0.016
	-	-		0.020	79.28 ± 1.35	
	-	-		0.010	21.66 ± 1.25	

*^a^* Trolox was used as a positive control for hypochlorous acid (HOCl) scavenging assay; whereas, quercetin, quercitrin, and epalrestat were used as positive controls for aldose reductase (AR) inhibition assay. *^b^* Data were presented as mean ± SD (*n* = 3). *^c^* Half-maximal inhibitory concentration (IC_50_). *^d^* Not applicable.

**Table 2 foods-10-01079-t002:** Partition coefficients (*K*_upper/lower_) of target compounds **1–7** in different biphasic solvent systems under acidic or basic conditions.

Solvent System (*v*/*v*)	Addition of Acid or Base	*K*_(upper/lower)_ Value						
		1	2	3	4	5	6	7
EtOAc/H_2_O, 1:1	208 mM formic acid	0.49	0.95	1.07	3.49	7.71	12.24	23.71
	30 mM ammonia	0.02	0.01	0.02	0.02	0.01	0.01	0.02
EtOAc/*n*-BuOH/H_2_O, 4:1:5	208 mM formic acid	1.36	2.43	3.05	4.86	14.23	22.17	27.56
	30 mM ammonia	0.00	0.01	0.01	0.02	0.01	0.01	0.03
EtOAc/*n*-BuOH/H_2_O, 3:2:5	208 mM formic acid	2.03	5.48	4.76	16.14	35.91	41.13	26.44
	30 mM ammonia	0.05	0.03	0.05	0.03	0.01	0.05	0.22
EtOAc/*n*-BuOH/H_2_O, 2:3:5	208 mM formic acid	2.49	6.86	5.22	20.24	38.56	45.95	29.51
	30 mM ammonia	0.01	0.01	0.01	0.04	0.12	0.13	0.40

Note: *n*-BuOH has been saturated using distilled water (H_2_O) before preparing the solvent systems.

**Table 3 foods-10-01079-t003:** Modification of the solvent system by adding acetic acid.

HSCCC System (*v*/*v*)	*K*_(upper/lower)_ Value	
	1	3
*n*-BuOH/H_2_O, 1:1	0.04	0.13
*n*-BuOH/H_2_O, 1:1 + 8.7 mM acetic acid	0.73	1.57

Note: *n*-BuOH was saturated using distilled water (H_2_O) before preparing the solvent systems.

**Table 4 foods-10-01079-t004:** Antioxidant activity of the seven target components from the 70% MeOH root extract of *V. rigida* using DPPH, ABTS, and ORAC assays.

Sample	TE (µmol Trolox Equivalents Per µmol Compound)		
	DPPH	ABTS	ORAC
3-CQA (1)	0.26 ± 0.00 *^e^*	0.28 ± 0.02 *^f^*	1.47 ± 0.10 *^c^*
4-CQA (3)	0.83 ± 0.05 *^c^*	0.84 ± 0.02 *^d^*	3.54 ± 0.11 *^a^*
5-CQA (2)	0.49 ± 0.01 *^d^*	0.58 ± 0.03 *^e^*	2.21 ± 0.04 *^ab^*
3,4-diCQA (4)	1.09 ± 0.09 *^b^*	1.06 ± 0.06 *^c^*	3.26 ± 0.10 *^a^*
3,5-diCQA (5)	0.86 ± 0.08 *^c^*	0.89 ± 0.05 *^d^*	2.13 ± 0.07 *^b^*
4,5-diCQA (6)	1.23 ± 0.07 *^a^*	1.22 ± 0.03 *^b^*	3.70 ± 0.34 *^a^*
3,4,5-triCQA (7)	1.28 ± 0.04 *^a^*	1.31 ± 0.05 *^a^*	1.95 ± 0.15 *^b^*

Note: the antioxidant activity is expressed as Trolox equivalent antioxidant capacity (TE) (µmol Trolox equivalents/µmol compound). Data was presented as mean ± SD (*n* ≥ 3). The statistical analysis was carried out using one-way ANOVA for DPPH assay (LSD test), ABTs assay (LSD test), and ORAC assay (tamhane T2 Test). Different lowercase letters mean significant difference (*p* < 0.05).

**Table 5 foods-10-01079-t005:** Hypochlorous acid scavenging activity and AR inhibitory activity of the components separated from the 70% MeOH root extract of *V. rigida*.

Sample *^a^*	HOCl Scavenging Activity			AR inhibitory Activity		
	Concentration (µM)	Inhibition (%) *^b^*	IC_50_ *^c^* (µM)	Concentration (µM)	Inhibition (%) *^b^*	IC_50_ (µM)
3-CQA (1)	100	70.79 ± 2.40	68.73	6.25	60.06 ± 1.22	7.80
	50	38.98 ± 1.33		3.125	45.84 ± 1.75	
	25	18.77 ± 0.92		1.5625	29.35 ± 3.40	
4-CQA (3)	100	84.70 ± 2.82	58.13	6.25	54.23 ± 2.18	4.83
	50	47.18 ± 1.01		3.125	42.88 ± 3.09	
	25	18.64 ± 1.72		1.5625	30.95 ± 2.76	
5-CQA (2)	100	91.90 ± 0.55	58.80	1.563	63.81 ± 2.55	0.91
	50	41.27 ± 1.08		0.781	44.21 ± 1.17	
	25	15.40 ± 1.47		0.391	30.80 ± 2.18	
3,4-diCQA (4)	12.5	59.15 ± 1.19	5.78	0.391	68.67 ± 0.68	0.22
	6.25	50.28 ± 0.19		0.195	50.40 ± 3.34	
	3.125	46.68 ± 2.87		0.098	18.86 ± 2.43	
3,5-diCQA (5)	12.5	62.86 ± 1.23	2.93	0.391	75.60 ± 1.33	0.15
	6.25	56.13 ± 3.01		0.195	62.33 ± 1.33	
	3.125	49.56 ± 1.23		0.098	35.37 ± 1.33	
4,5-diCQA (6)	12.5	65.80 ± 4.06	4.61	0.391	65.87 ± 2.21	0.23
	6.25	51.23 ± 2.14		0.195	44.36 ± 4.45	
	3.125	48.80 ± 5.62		0.098	23.73 ± 2.70	
3,4,5-triCQA (7)	25	84.32 ± 3.39	11.68	0.781	69.85 ± 0.77	0.40
	12.5	52.93 ± 1.02		0.391	49.73 ± 0.94	
	6.25	35.39 ± 1.17		0.195	27.56 ± 2.43	
Acacetin (8)	- *^d^*	-	-	100	36.30 ± 3.78	-
	-	-		50	24.00 ± 0.63	
	-	-		25	12.30 ± 4.22	
Trolox	100	97.07 ± 5.45	34.03	-	-	-
	50	66.53 ± 2.54		-	-	
	25	40.76 ± 0.39		-	-	
Quercetin	-	-	-	25	67.05 ± 3.22	15.01
	-	-		12.5	45.98 ± 4.49	
	-	-		6.25	34.78 ± 0.51	
Quercitrin	-	-	-	0.25	74.57 ± 1.11	0.10
	-	-		0.125	62.33 ± 0.44	
	-	-		0.063	32.57 ± 2.70	
Epalrestat	-	-	-	0.125	91.81 ± 1.84	0.05
	-	-		0.063	79.28 ± 1.35	
	-	-		0.031	21.66 ± 1.25	

*^a^* Compounds **1–8** were separated from the 70% MeOH root extract of *V. rigida*. Particularly, compounds **1**–**7** were the target compounds screened by offline DPPH-HPLC and ultrafiltration-HPLC; Trolox was used as a positive control for hypochlorous acid (HOCl) scavenging assay, whereas quercetin, quercitrin, and epalrestat were used as positive controls for aldose reductase (AR) inhibition assay. *^b^* Data was presented as mean ± SD (*n* = 3). *^c^* Half-maximal inhibitory concentration (IC_50_). *^d^* Not measured or not applicable.

**Table 6 foods-10-01079-t006:** Calibration curve, LOD, LOQ, repeatability, and reproducibility test of the three major compounds in the 70% MeOH root extract of *V. rigida*.

Parameter		Compound		
		3,5-diCQA (5)	4,5-diCQA (6)	Acacetin (8)
Calibration curve (*n* = 3), *r*^2^		y = 0.184x − 0.0692, *r*^2^ = 0.9996	y = 0.1297x − 0.4051, *r*^2^ = 0.9991	y = 0.4122x + 0.9041, *r*^2^ = 0.9987
Linear range (µg/mL)		12.50–400.00	12.50–400.00	6.25–400.00
LOD (µg/mL, S/N = 3)		0.05	0.06	0.20
LOQ (µg/mL, S/N = 10)		0.17	0.20	0.65
Precision on the day in relative standard deviation, % (*n* = 6)	50.00 µg/mL	8.70	6.64	6.45
	200.00 µg/mL	7.60	5.80	6.24
Precision between days in relative standard deviation, % (*n* = 3)	50.00 µg/mL	3.30	4.73	4.62
	200.00 µg/mL	7.40	4.10	3.57

Note: LOD and LOQ are the abbreviations of the limit of detection and the limit of quantification, respectively.

**Table 7 foods-10-01079-t007:** The spike recovery test and content of the three major compounds in the 70% MeOH root extract of *V. rigida*.

Analyte	*C*_0_ (µg/mL)	*V*_o_ (mL)	*C*_1_ (µg/mL)	*V*_1_ (mL)	*C*_2_ (µg/mL)	*V*_2_ (mL)	Recovery (%)	Content (mg/g)
3,5-diCQA (5)	100	0.2	66.47 ± 3.95	0.2	82.89 ± 3.80	0.4	99.32 ± 7.60	66.47 ± 3.95
	300	0.2	66.47 ± 3.95	0.2	184.62 ± 6.54	0.4	100.93 ± 4.36	
4,5-diCQA (6)	100	0.2	112.95 ± 2.63	0.2	107.08 ± 3.44	0.4	101.21 ± 6.88	112.95 ± 2.63
	300	0.2	112.95 ± 2.63	0.2	206.16 ± 5.85	0.4	99.79 ± 3.90	
Acacetin (8)	100	0.2	23.26 ± 0.85	0.2	60.69 ± 2.83	0.4	98.11 ± 5.66	23.26 ± 0.85
	300	0.2	23.26 ± 0.85	0.2	155.63 ± 6.20	0.4	96.00 ± 4.13	

Note: spike recovery (%) = (*C*_2_ × *V*_2_ − *C*_1_ × *V*_1_)/(*C*_0_ × *V*_0_) × 100%, where *C*_0_, *C*_1_, and *C*_2_ are the concentrations of each compound tested in the standard solution, extract solution (the extract was prepared as 1.00 mg/mL), and the spiked solution, respectively. *C*_1_ and *C*_2_ are calculated from the corresponding calibration curve of each compound; *V*_0_, *V*_1_, and *V*_2_ are the volumes of the standard solution used for sample spiking, the extract solution used for sample spiking, and the spiked sample solution, respectively. The content (µg/mg = mg/g) of each component in the extract was measured as *C*_1_ (µg/mL)/extract concentration (1.00 mg/mL).

## Data Availability

Data are contained within the article or Supplementary data.

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
