# Peer review of "Valeriana rigida* Ruiz & Pav. Root Extract: A New Source of Caffeoylquinic Acids with Antioxidant and Aldose Reductase Inhibitory Activities"

_foods, 2021, doi:10.3390/foods10051079_

Round 1

Reviewer 1 Report

Please address the following suggestions to improve the present manuscript.

  1. INTRODUCTION.

In general, the introduction laks in V. rigida’s health benefits, mainly related with diabetes. Also, the relevance of certain bioactive compounds, i.e. quercetin and quercitrin, against diabetes, is not clear enough, in contrast they are used as a AR inhibitors all over the text.

Line 54. It seems like the authors mainly focus aldose reductase inhibition and its relation with natural compounds/extracts. Please address more details about this in the introduction.

Lines 63-66. The paragraph, that starts in the line 63 and ends in the line 66 (...AR inhibitors that may 65 promote health and reveal antidiabetic properties. ), doesn't make sense in the introduction section. You have to talk about the state of the art of V. rigida, instead of the results in the present study. You can mention the bioactivity known of V. rigida, mainly related with the prevention of diabetes mellitus, even if it's limited.

  1. MATERIALs & METHODS

Line 334. The authors don’t refer from where the standard solutions are.

  1. RESULTS

In the 3.1 section, is table 1 needed? data is already presented in text.

In the 3.5 section, you don’t have to repeat the numeric result, because it can be observed in the Table 5 an Table 6.

In the 3.6 section, it’s not clear why the authors used the 254 nm signal to quantify the acacetin compound. This compound is a flavonoid which its maximum UV absorption is around 340-350 nm of absorbance.

  1. DISCUSSION

Line 626- The authors refer to ‘acacetin’ as compound number 7, but in other sections they refer as compound number 8. Please correct.

Line 631- Correct the mistake in this line, because 3,5-diCQA appears twice. (…3,4-diCQA and 3,5-diCQA appeared to exert higher activity than 3,5-diCQA against…).

Author Response

Reviewer 1: Please address the following suggestions to improve the present manuscript.

Response: Thank you very much for your valuable suggestions, we have revised the manuscript to improve its quality as you suggested.

Comment 1: INTRODUCTION part. In general, the introduction laks in V. rigida’s health benefits, mainly related with diabetes. Also, the relevance of certain bioactive compounds, i.e. quercetin and quercitrin, against diabetes, is not clear enough, in contrast they are used as a AR inhibitors all over the text.

Response: Thank you very much for your valuable suggestions. The relevant references were added to support curcuminoids, resveratrol, quercitrin, quercetin, and chlorogenic acid as potential anti-diabetic agents through inhibition of ROS/oxidative stress or aldose reductase in lines 61-63 in this revised manuscript. However, no scientific publications could be found to report V. rigida’s anti-diabetic effects. Whereas some species of Valeriana L., such as V. officinalis, V. dioscoridis, and V. wallichii have previously shown anti-diabetic potential, and this information has been indicated in lines 71-73 in this revised version. In addition, the newly added references were marked using red color.

Comment 2: INTRODUCTION part. Line 54. It seems like the authors mainly focus aldose reductase inhibition and its relation with natural compounds/extracts. Please address more details about this in the introduction.

Response: Thank you very much for your critical comments. we have listed some representative natural aldose reductase inhibitors, e.g., quercitrin, quercetin, and chlorogenic acid in lines 62-63 in this revised version.

Comment 3: INTRODUCTION part. Lines 63-66. The paragraph, that starts in the line 63 and ends in the line 66 (...AR inhibitors that may 65 promote health and reveal antidiabetic properties. ), doesn't make sense in the introduction section. You have to talk about the state of the art of V. rigida, instead of the results in the present study. You can mention the bioactivity known of V. rigida, mainly related with the prevention of diabetes mellitus, even if it's limited.

Response: thank you very much for your critical comments. the information (lines 63-66 in the initial manuscript) reporting V. rigida to have antioxidant and AR inhibitory activities has been deleted as you suggested. However, the scientific publications reporting anti-diabetic potential on V. rigida is so limited that no relevant reference can be found. We have indicated this background in lines 71-73 in this revised manuscript. Moreover, the newly added references were marked using red color.

Comment 4: MATERIALs & METHODS part. Line 334. The authors don’t refer from where the standard solutions are.

Response: thank you very much for your careful checking. The standard solutions were prepared using the purified compounds in this study, which has been newly indicated in this revised version in lines 336-338.

Comment 5: Results part. In the 3.1 section, is table 1 needed? data is already presented in text.

Response: thank you very much for your kind suggestion. The Table 1 has been moved to supplementary data (Table S1) in the revised version. Moreover, all the table numbers have been updated in lines 375 (Table S1), 377 (Table 1), 381 (Table 1), 386 (Table 1), 441 (Table S2), 449 (Table S2), 458 (Table 2), 473 (Table 2), 501 (Table 3), 508 (Table 3), 523 (Table S3), 537 (Table 4), 538 (Table 5), 540 (Tables 4 and 5), 541 (Table 4), 545 (Table 4 and Table 5), 547 (Table 2), 548 (Table 5), 554 (Table 5), 562 (Table 4), 569 (Table 5), 583 (Table 6), 587 (Table 6), 590 (Table 6), 591 (Table 6), 595 (Table 7), 598 (Table 7), 601 (Table 7), 619 (Table S1 and Table 1), 622 (Tables 4 and 5), 625 (Table 7), 627 (Tables 4 and 5), 630 (Table 4), 634 (Table 4), 636 (Tables 4 and 5), 641 (Table 5), 644 (Table 5), 666 (Tables 4 and 5), 668 (Table 5), 715 (Table S1), 717 (Table S2), 718 (Table S3), and the table numbers in supplementary data.

Comment 6: Results part. In the 3.5 section, you don’t have to repeat the numeric result, because it can be observed in the Table 5 and Table 6.

Response: thank you very much for your good suggestions. the repeated numeric results have been removed in section 3.5 (lines 539-561) in this revised version.

Comment 7: Results part. In the 3.6 section, it’s not clear why the authors used the 254 nm signal to quantify the acacetin compound. This compound is a flavonoid which its maximum UV absorption is around 340-350 nm of absorbance.

Response: thank you very much for your good question. To simplify the HPLC quantification process, we used the sample HPLC condition as the DPPH-/ultrafiltration-HPLC offline did, which has been indicated in lines 333-335 in the revised version.

Comment 8: Discussion part. Line 626- The authors refer to ‘acacetin’ as compound number 7, but in other sections they refer as compound number 8. Please correct.

Response: thank you very much to point out our mistake. The compound number of acacetin has been corrected to 8 in line 624 in this revised version.

Comment 9: Discussion part. Line 631- Correct the mistake in this line, because 3,5-diCQA appears twice. (…3,4-diCQA and 3,5-diCQA appeared to exert higher activity than 3,5-diCQA against…).

Response: thank you very much to point out our mistake. the correct one should be “whereas 3,4-diCQA and 4,5-diCQA appeared to exert higher activity than 3,5-diCQA against DPPH, ABTS, and peroxyl (ORAC assay) radicals”, which has been corrected in lines 629-630 in this revised version.

Reviewer 2 Report

The work is interesting and the manuscript is well written, structured and the results may have practical applications industry as well as for the researchers in the field but needs minor revisions to be improved.

Comments to the Author

This study aimed to broaden new natural compounds with antioxidant and AR inhibitory activity from root extract of V. rigida. The topic is of an importance to the food and medical science community and related area. The manuscript is well written and structured; the results may have practical applications for the food and pharmaceutical industry as well as for the researchers in the field. The work is interesting but needs minor revisions to be improved.

Introduction:

Lines 42-57: I suggest that the authors begin introduction to the contrary: first importance of natural products as sources of antioxidants and AR inhibition and after the relationship of AR inhibition with diabetes mellitus because of the aim of this word is the compounds with those activities from a natural product.

Lines 63-68: this information is results. If it is refereed at bibliography, authors have to include same reference.

Material and Methods

For a better compression, I recommended modified section 2.1 only as Reagents and section 2.2 Vegetal material and Preparation of plant extract.

Author Response

Reviewer 2.

Overall comment: The work is interesting and the manuscript is well written, structured and the results may have practical applications industry as well as for the researchers in the field but needs minor revisions to be improved.

Response: thank you very much for your positive comments, we have further revised this manuscript as you and other reviewers’ valuable suggestions.

Comments to the Author: This study aimed to broaden new natural compounds with antioxidant and AR inhibitory activity from root extract of V. rigida. The topic is of an importance to the food and medical science community and related area. The manuscript is well written and structured; the results may have practical applications for the food and pharmaceutical industry as well as for the researchers in the field. The work is interesting but needs minor revisions to be improved.

Response: thank you very much for your positive comments and kind suggestions, we have further improved the manuscript as you suggested.

 Specific comments,

Comment 1: Introduction part. Lines 42-57: I suggest that the authors begin introduction to the contrary: first importance of natural products as sources of antioxidants and AR inhibition and after the relationship of AR inhibition with diabetes mellitus because of the aim of this word is the compounds with those activities from a natural product.

 Response: thank you very much for your suggestion. We have re-written this paragraph as you suggested in lines 42-64 in this revised version. Moreover, the newly added references were marked using red color.

Comment 2: Introduction part. Lines 63-68: this information is results. If it is refereed at bibliography, authors have to include same reference.

 Response: thank you very much for your critical comment. the antioxidant and AR inhibitory activities of V. rigida are reported for the first time in this study. therefore, the lines 63-68 in the initial manuscript were deleted. In addition, the relevant background has been re-written in lines 70-73 in the revised version. In addition, the newly added references were marked using red color.

Comment 3: Material and Methods part. For a better compression, I recommended modified section 2.1 only as Reagents and section 2.2 Vegetal material and Preparation of plant extract.

Response: thank you very much for your good suggestion. These two sections have been modified as you suggested in lines 96 and 113 in this revised version.

Reviewer 3 Report

The authors prepared root extract from Valeriana rigida Ruiz and Pav. (V. rigida) using 70% MeOH and tested it for its antioxidant and aldose reductase (AR) inhibitory potential. Offline 2,2-diphenyl-1-picryl-hydrazyl-hydrate (DPPH)-/ultrafiltration-high performance liquid chromatography (HPLC) and high-speed counter-current chromatography (HSCCC)/pH-zone-refining counter-current chromatography (pH-zone-refining CCC) techniques were utilized to identify and separate seven different caffeoylquinic acids. These were found to demonstrate considerable antioxidant and AR inhibitory potential in relevant assays. Based on this work, the authors suggest that V. rigida can be used as a botanical source of functional ingredients against diabetes.

Specific suggestions for the improvement of the manuscript are as following:

Page 2

Line 63-65: This discussion doesn’t belong to the “Introduction” section. Please consider deleting this statement and covering it later in “Results” section.

Page 3

Line 116: filtered…

Line 120: a STH585…

Line 137: the results were…

Line 140: % inhibition =

Page 4

Line 154: the results were presented…

Line 160-161: Please specify the full form of ORAC assay

Line 193: % inhibition

Page 5

Line 200: (precooled at -70 °C)

Line 202: using a 5417R centrifuge…

Line 218: results were…

Line 221: % inhibition

Line 240-241: mL (please be consistent)

Page 6

Line 262: on the principle…

Line 268-272: mL

Page 8

Line 349: % Spike recovery

Line 376: may be used as new sources…

Page 17

Line 605: spike recovery (%)…

Line 635: of the major components were…

Page 18

Line 673: which also functioned well…

Author Response

Reviewer 3,

Overall comment: The authors prepared root extract from Valeriana rigida Ruiz and Pav. (V. rigida) using 70% MeOH and tested it for its antioxidant and aldose reductase (AR) inhibitory potential. Offline 2,2-diphenyl-1-picryl-hydrazyl-hydrate (DPPH)-/ultrafiltration-high performance liquid chromatography (HPLC) and high-speed counter-current chromatography (HSCCC)/pH-zone-refining counter-current chromatography (pH-zone-refining CCC) techniques were utilized to identify and separate seven different caffeoylquinic acids. These were found to demonstrate considerable antioxidant and AR inhibitory potential in relevant assays. Based on this work, the authors suggest that V. rigida can be used as a botanical source of functional ingredients against diabetes.

Specific suggestions for the improvement of the manuscript are as following:

Response: thank you very much for your careful reading and kind suggestions. We have corrected all the errors as you suggested. Moreover, we have checked the manuscript carefully and found one mistake about the quantity of the separated compound 5 (300.2 mg, it should be 146.5 mg) and 6 (146.5 mg, it should be 300.2 mg), and this has been corrected in lines 479 and 697 in the revised version.

Specific suggestions,

Comment 1: Page 2 Line 63-65: This discussion doesn’t belong to the “Introduction” section. Please consider deleting this statement and covering it later in “Results” section.

Response: Thank you very much for your kind suggestion. The sentence “The 70% MeOH root extract of V. rigida used in our study exhibited strong antioxidant and AR inhibitory activities, thereby indicating that the root extract and the phytochemicals that it contains may act as new sources of antioxidants and AR inhibitors that may promote health and reveal antidiabetic properties.” has been deleted, and the relevant information has been covered in lines 381-384 in the revised manuscript. Moreover, the newly added references were marked using red color.

Comment 2: Page 3. Line 116: filtered…

Response: Thank you very much for your kind suggestion. The “filtrate” has been replaced with “filtered” in line 121 in this revised version.

Comment 3: Page 3. Line 120: a STH585…

Response: Thank you for your kind suggestion. The “an STH858” has been corrected to “a STH585” in line 125 in this revised version.

Comment 4: Page 3. Line 137: the results were…

Response: Thanks a lot for your good suggestion. “the result was” has been corrected to “the results were” in line 142 in this revised version.

Comment 5: Page 3. Line 140: % inhibition =

Response: Thanks a lot for your good suggestion. The “%inhibition =” was corrected to “% inhibition =” in line 145 in this revised version.

Comment 6: Page 4. Line 154: the results were presented…

Response: Thanks a lot for your good suggestion. “the result was” has been corrected to “the results were” in line 159 in this revised version.

Comment 7: Page 4. Line 160-161: Please specify the full form of ORAC assay

Response: Thanks a lot for your good suggestion The full form of ORAC assay was added in lines 165-166 in this revised version.

Comment 8: Page 4. Line 193: % inhibition

Response: Thanks a lot for your good suggestion. The “%inhibition =” was corrected to “% inhibition =” in line 198 in this revised version.

Comment 9: Page 5. Line 200: (precooled at -70 °C)

Response: Thanks a lot for your good suggestion. The “(precooled -at -70 °C)” was corrected to “(precooled at -70 °C)” in line 205 in this revised version.

Comment 10: Page 5. Line 202: using a 5417R centrifuge…

Response: Thanks a lot for your good suggestion. The “using a Centrifuge 5417R centrifuge” has been corrected to “using a 5417R centrifuge” in line 207 in this revised version.

Comment 11: Page 5. Line 218: results were…

Response: Thanks a lot for your good suggestion. “the result was” has been corrected to “the results were” in line 223 in this revised version.

Comment 12: Page 5. Line 221: % inhibition

Response: Thanks a lot for your good suggestion. The “%inhibition =” was corrected to “% inhibition =” in line 226 in this revised version.

Comment 13: Page 5. Line 240-241: mL (please be consistent)

Response: Thanks a lot for your good suggestion. “ml” has been corrected to “mL” in lines 245 and 246.

Comment 14: Page 6. Line 262: on the principle…

Response: Thanks a lot for your good suggestion. “as the principle” has been corrected to “on the principle” in line 267 in this revised version.

Comment 15: Page 6. Line 268-272: mL

Response: Thanks a lot for your good suggestion. “ml” has been corrected to “mL” in lines 273 and 277.

Comment 16: Page 8. Line 349: % Spike recovery

Response: Thanks a lot for your good suggestion. The “% Spike recovery =” was corrected to “% Spike recovery =” in line 356 in this revised version.

Comment 17: Page 8. Line 376: may be used as new sources…

Response: Thanks a lot for your good suggestion. “maybe act as new sources” has been replaced with “may be used as new sources” in line 383 in this revised version.

Comment 18: Page 17. Line 605: spike recovery (%)…

Response: Thanks a lot for your good suggestion. “spike recover (%)” was corrected to “spike recovery (%)” in line 603 in this revised version.

Comment 19: Page 17. Line 625: of the major components were…

Response: Thanks a lot for your good suggestion. “the contents (mg compound/g extract) of the major components 3,5-diCQA (5), 4,5-di-CQA (6), and acacetin (8) are” has been corrected to “the contents (mg compound/g extract) of the major components 3,5-diCQA (5), 4,5-di-CQA (6), and acacetin (8) were” in line 624.

Comment 20: Page 18. Line 673: which also functioned well…

Response: Thanks a lot for your good suggestion. “which also functions appropriately” has been replaced with “which also functioned well” in line 671 in this revised version.
